# A Machine-Learning-Algorithm-Assisted Intelligent System for Real-Time Wireless Respiratory Monitoring

**Chi Zhang [†], Lei Zhang \*,[†], Yu Tian, Bo Bao and Dachao Li \***

State Key Laboratory of Precision Measuring Technology and Instruments, Tianjin University, Tianjin 300072, China
* Correspondence: zhangleitd@tju.edu.cn (L.Z.); dchli@tju.edu.cn (D.L.); Tel.: +86-18583636676 (L.Z.)
† These authors contributed equally to this work.

**Abstract:** Respiratory signals are basic indicators of human life and health that are used as effective biomarkers to detect respiratory diseases in clinics, including cardiopulmonary function, breathing disorders, and breathing system infections. Therefore, it is necessary to continuously measure respiratory signals. However, there is still a lack of effective portable electronic devices designed to meet the needs of daily respiratory monitoring. This study presents an intelligent, portable, and wireless respiratory monitoring system for real-time evaluation of human respiratory behaviors. The system consists of a triboelectric respiratory sensor; circuit board hardware for data acquisition, preprocessing, and wireless transmission; a machine learning algorithm for enhancing recognition accuracy; and a mobile terminal app. The triboelectric sensor—fabricated by the screen-printing method—is lightweight, non-invasive, and biocompatible. It provides a clear response to the frequency and intensity of respiratory airflow. The portable circuit board is reusable and cost-effective. The decision tree model algorithm is used to identify the respiratory signals with an average accuracy of 97.2%. The real-time signal and statistical results can be uploaded to a server network and displayed on various mobile terminals for body health warnings and advice. This work promotes the development of wearable health monitoring systems.

**Keywords:** respiratory monitoring; triboelectric sensor; decision tree model; machine learning algorithm; intelligent mask



## 1. Introduction

Respiration is a fundamental indicator of life throughout human life, including the cycle of inhalation and exhalation [1,2]. It can be used clinically as an effective biomarker for detecting respiratory disease—the second leading cause of death, disability, and morbidity, after cardiovascular disease—and for early screening of chronic obstructive pulmonary disease, bronchial asthma, sleep apnea, and respiratory infection [3,4]. For example, COVID-19 and influenza virus H1N1 are intimately associated with the respiratory system, and most patients experience long-term respiratory discomfort during the rehabilitation process [5,6], including coughing, severe coughing, sneezing, rapid breathing, etc. At present, the clinical diagnosis of respiratory diseases is based on medical history, symptoms, and stage treatment effects. Therefore, based on the characteristics of respiratory system diseases and corresponding symptoms, it is beneficial for patients to continuously monitor respiratory behavior and extract clinically relevant parameters to accurately describe their respiratory status [7]. Continuous and real-time respiratory monitoring using portable equipment is an effective method for tracking lung function, quantitatively changing treatment plans and evaluating treatment effects, and prognosis [8]. Therefore, intelligent respiratory monitoring systems that are portable, low-cost, and intelligent are extremely beneficial for personal health via real-time monitoring of human respiratory physiological status.

In recent years, portable respiration sensing devices have been widely studied and are seen as effective means of monitoring respiratory health [9,10]. For example, Zhang et al. [11]



developed a biodegradable respiratory sensor to distinguish between healthy and asthmatic, bronchitis, and chronic obstructive pulmonary disease states. Chen et al. [12] reported a respiratory monitoring system that can simultaneously sample respiratory and temperature signals using both capacitive and resistive sensors. Fang et al. [13] integrated a five-channel triboelectric sensing network on a mask, which can overcome environmental interference due to different facial contours and achieve a recognition accuracy of up to 100%. Among these sensors, the triboelectric sensor, based on the coupling effect of triboelectrification and electrostatic induction, has great potential among wearable electronics due to its simple structure, light weight, and self-powered characteristics [14–19]. However, there are still some challenges to overcome in order to gain a portable and sufficiently smart respiratory monitoring system. These challenges include making respiration sensors highly portable and comfortable for the human face, achieving higher accuracy to recognize complex respiratory behaviors, and improving system-level and multi-terminal respiratory monitoring applications.

In this study, we report a portable, algorithm-assisted, and wireless respiratory monitoring system for real-time evaluation of human respiratory behaviors. The triboelectric respiratory sensor was directly screen-printed onto a commercial mask with a thickness of 50 μm and a weight of 0.05 g to capture respiratory signals (respiratory intensity and respiratory frequency) and showed high wearability and compatibility in daily use. A portable integrated circuit board was fabricated for signal acquisition, preprocessing, and wireless transmission, which can be reused, and only the low-cost respiratory sensing mask would need to be replaced in long-term use, significantly reducing the cost of the whole system. A decision tree model algorithm is used to categorize and quantify respiratory signals. A total of 12 features are extracted via a trainable classifier, and an average accuracy of 97.2% has been validated for five types of respiratory status. The real-time signals and statistical results of the respiratory status of the user can be uploaded to a server network and displayed in various mobile terminals for future body health warnings and advice.

## 2. Materials and Methods

### 2.1. Materials and Fabrication of the Triboelectric Respiratory Sensor

A commercial screen printer (PHP-B Series, Shanghai Hoting Screen Printing Equipment Co., Ltd., Shanghai, China.) was used to manufacture the triboelectric respiratory sensors. Ag ink was purchased from Qingdao Nano Print Materials Technology Co., Ltd, Qingdao, China. Nylon ink was purchased from Hangzhou MIHE Trading Co., Ltd., Hangzhou, China. The Ag and nylon inks were directly used as received without further purification. PTFE film was purchased from Dongguan Hongfu Insulating Material Co., Ltd., Dongguan, China.

Subsequently, the nylon, Ag, and nylon inks were printed layer by layer on the mask textile substrate and dried completely at 120 °C for 30 min. The PTFE film was fixed above the printed nylon ink to form a micro-curved surface. A structural diagram is shown in Figure 1b.

### 2.2. Fabrication of Intelligent Wireless Respiratory Monitoring System

A commercial mask made of polypropylene spun-bonded non-woven fabric with a double breathing valve was bought. The self-powered triboelectric respiratory sensor was directly screen-printed onto the innermost non-woven textile surface of the mask to form an intelligent mask for real-time respiratory monitoring. A circuit board was built for data acquisition with a sampling frequency of 1000 point/s and wireless signal transmission, which includes a low-pass multi-channel filter, an AD623 amplifier [20], an ADS1256 analog-to-digital converter [21], and a BL600-SA-32 Bluetooth module [22]. A decision tree model of the machine learning algorithm was used to enhance the recognition accuracy of respiratory behaviors. An app was programmed for mobile phone terminals. The real-time respiratory signals and statistical results for various respiration behaviors were displayed in the app user interface.

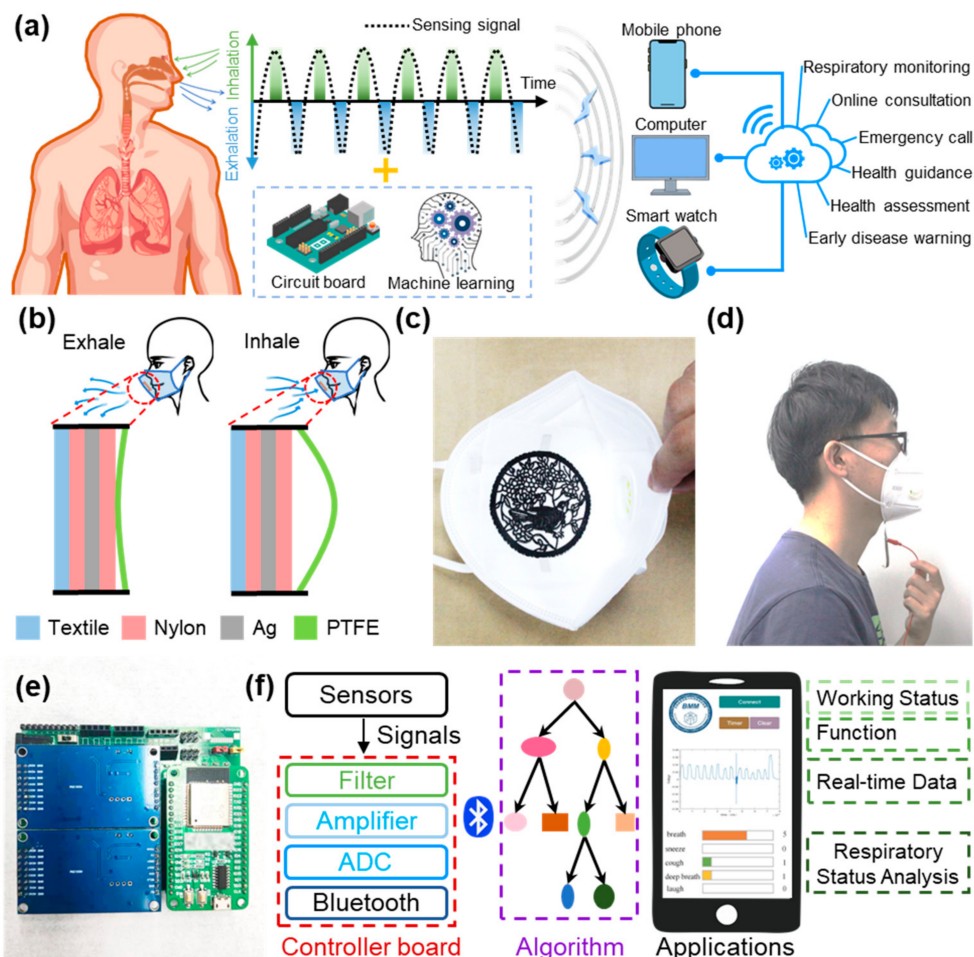

**Figure 1.** The main idea of the wireless intelligent respiratory monitoring system. (**a**) Schematic diagram of the wearable respiratory monitoring system and its potential expandable medical services. (**b**) The structural changes of the triboelectric respiratory sensor during the respiratory cycle, that is, the contact between the PTFE and nylon during the expiratory process, and the separation of the PTFE and nylon during the inspiratory process. (**c**,**d**) Photograph of the triboelectric respiratory sensor assembled with the mask and worn on a volunteer's face. (**e**) Photograph of self-built wireless and portable data-acquisition circuit for the triboelectric respiratory sensor. (**f**) Logical flow chart of the wireless respiratory monitoring system, including respiratory signal acquisition, preliminary processing, wireless transmission, learning and testing of the algorithm, and display of respiratory results in the app user interface.

### 2.3. Data Processing and Analysis by a Decision Tree Model Algorithm

The five types of typical respiratory behaviors (normal breathing, deep breathing, sneezing, coughing, and laughing) were collected by the self-built circuit board and then pretreated by filtering and smoothing.

The data analysis process was mainly based on the data analysis tools of wavelet transform and Fourier transform. Wavelet transform was utilized to analyze transient non-stationary signals and transform the original signals into time–frequency signals, which can fully highlight the temporal distributional differences of various low-frequency respiratory signals. Fourier transform transforms the time-domain signal into the frequency-domain signal, which can be used to intuitively distinguish the frequency concentration and fluctuation range and is used to judge the changes in respiratory behavior that lead to this result. Based on the characteristics of the above three kinds of spectra, a total of 12 features were extracted, including mean, variance, standard deviation, root mean square and kurtosis from the original time domain signal; pulse factor, waveform factor, peak

factor, and skewness from the time–frequency domain; and unbiased estimation, coefficient of variation, and edge factor from the frequency domain.

Next, the decision tree algorithm was used to build a classification model based on the above features to improve the recognition accuracy of respiratory behavior. The total number of samples in the training data set was 1580, and the number of samples for each respiratory behavior was 316. Finally, the test data set including 395 data points was used to evaluate the training results.

### 2.4. Characterization and Electrical Measurements

The output voltages, currents, and charges of the self-powered triboelectric respiratory sensors were measured by Keithley 6514. Data were recorded using self-programmed software written in LabView. The periodic respiratory process was simulated by a linear motor (Linmot E1100, LinMot, Inc., Elkhorn, WI, USA), and the constant pressure value was monitored by a Vernier double-range force sensor (Vernier Science Education, Beaverton, OR, USA).

### 3. Results

#### 3.1. Engineering Mechanism of the Intelligent Respiratory Monitoring System

As shown in Figure 1a, the typical characteristic of physiological respiration is the cycle of inhalation and exhalation of air, which directly reflects respiratory status. Our main idea was to prepare an intelligent respiratory monitoring system which included a portable respiratory sensor to monitor the cycle of air inhalation and exhalation; a circuit board for real-time data acquisition, preprocessing, and wireless transmission; and a machine learning algorithm to enhance the recognition accuracy of respiratory behaviors. Finally, all test data were to be processed and presented in the user interfaces of various terminals to display real-time respiratory results and possible health-status alarms.

Based on the above considerations, a respiratory sensor combined with a daily mask is an effective tool for daily respiratory monitoring. However, disposable masks are limited in their use time, and the average person cannot wear a mask for more than 8 h. In the COVID-19 pandemic environment, doctors suggested that the cumulative use time of a mask should not exceed 4 h and insisted that it should not be reused. Therefore, the wearing experience and production cost of the respiratory sensor are important factors that must be considered.

As shown in Figure 1b,c, a triboelectric respiratory sensor with a single electrode mode was utilized and fabricated by directly screen-printing biocompatible nylon and Ag inks on a daily mask. In the commercial production process of daily masks, screen printing is a mature customized processing technology, and this method is commonly used to print logos, models, and patterns on clothes. The nylon/Ag/nylon inks were layer-by-layer screen-printed onto the mask (Figure S1), then the PTFE film was tailored and fixed above the nylon layer with a small curvature. The thickness of the upper PTFE layer was 20 μm, and the maximum distance between the PTFE and nylon was 1 mm. A surface SEM image of the nylon and Ag ink layers is shown in Figure S2. During the exhalation and inhalation process, the airflow pushes the inner PTFE film so that it contacts or separates from the printed nylon layer. Based on the coupled effect of triboelectrification and electrostatic induction, this triboelectric respiratory sensor can directly sense the driving force of the respiratory airflow, which can provide intuitive respiratory information, with high-fidelity data, non-invasive acquisition, and a non-irritating interface. Figures S3 and S4 illustrate the thickness and the weight of our printed triboelectric respiratory sensor, which are only 50 μm and 0.05 g, respectively. The thicknesses of the top nylon layer, the Ag electrode, the bottom nylon layer, and the cotton textile are 23, 12, 16, and 500 μm, respectively. In practical use, with the merits of light weight, superior flexibility, and high integration with the mask, the triboelectric respiratory sensor can fit comfortably on the user's face (Figure 1d) and is fixed near the respiratory valve of the mask, so that it will not affect the filtering ability of the mask. The costs of each of the materials used in the $4 \times 4$ cm$^2$

respiratory sensor are listed in Table S1, and the total cost of replacing the respiratory monitoring mask is only 0.377 dollars, meaning that it does not carry an economic burden.

The output signal of the triboelectric respiratory sensor is recorded by a self-built portable data-acquisition circuit board, as shown in Figures 1e and S5. This circuit board can be reused, and only the low-cost respiratory sensing mask would need to be replaced in long-term use, which significantly reduces the cost of the whole system.

A logical flow chart for this wireless respiratory system is shown in Figure 1f. After the sampling, filtering, amplification, and transmission of the respiratory signals and machine learning, the dynamic respiratory curves and statistical data can be displayed in multi-terminal consumer electronics for real-time respiratory monitoring, health assessment, online consultation, and early disease warnings.

### 3.2. Working Mechanism and Output Performance of the Triboelectric Respiratory Sensor

Figure 2a shows the sensing mechanism of our triboelectric respiratory sensor [23–25]. The working process of the triboelectric respiratory sensor can be described as follows. First, in the original state, the electrification process occurs when the PTFE film contacts the nylon layer, and the same numbers of opposite-polarity charges are generated between the contact interfaces. As the most triboelectrically negative material, PTFE tends to gain electrons compared with nylon. Then, in the inhaling process, the PTFE film gradually separates from the nylon, and the potential difference increases, resulting in the instantaneous flow of electrons from the ground to the Ag electrode. In the exhalation process, the PTFE film approaches the nylon layer, and the potential difference decreases, leading to the electrons' instantaneous flow back to the ground until the two materials come into full contact. During periodic respiration, the PTFE film performs a periodic contact–separation process to generate an instantaneous AC current through an external circuit. To understand the potential distribution of the respiratory sensor more quantitatively, finite-element analysis was carried out using COMSOL software, as shown in Figure 2b. The potential balance is formed when the PTFE film is in full contact with the printed nylon layer. When the PTFE film is far away from the nylon layer, a potential difference is formed between the PTFE film with positive potential and the nylon layer with negative potential. The results of the finite-element analysis confirmed the mechanistic analysis.

To accurately evaluate the respiratory-sensing performance, a linear motor was used to drive the PTFE film to move reciprocally and simulate the respiratory process. As shown in Figure 2c, with an area of $4 \times 4$ cm$^2$, a frequency of 0.5 Hz, and a constant force of 1 N, our triboelectric respiratory sensor can output an open-circuit voltage ($V_{oc}$) of 8.3 V, a short-circuit current ($I_{sc}$) of 17.6 nA, and a short-circuit charge ($Q_{sc}$) of 2.5 nC. For the 0.5 Hz $V_{oc}$ signal, the response and recovery times of the respiratory sensor to the airflow are 0.25 s and 0.97 s, respectively, in the exhalation and inhalation process (Figure S6).

The electrical output of this triboelectric respiratory sensor is determined by the respiratory status, including two main factors, namely, respiratory intensity and respiratory frequency. Respiratory pressure and frequency in humans are generally lower than 1000 Pa and 3 Hz, respectively [13]. Firstly, Figure 2d shows that the output voltage of the triboelectric respiratory sensor increases with increasing pressure from 60 Pa to 3000 Pa and that the output voltage presents an approximately linear increase in the range of 60–1200 Pa, with a sensitivity of 0.0079 V/Pa. Secondly, as shown in Figures 2e and S7, with an operating frequency from 0.5 Hz to 3 Hz, the triboelectric respiratory sensor exhibits well-defined frequency characteristics, while the output voltage shows an indistinct difference and remains stable at 8.3 V.

Moreover, the electrical output can be easily regulated by the device area (Figure 2f). In particular, for patients with dyspnea or respiratory failure, the sensing area can be appropriately increased to enhance the signal-to-noise ratio to obtain clearer respiratory signals by means of the customization of screen-printing methods. Change in temperature has no effect on the respiratory sensor, which ensures the ability of the sensor to continue to be used in a changing environment (Figure S8). Durability is another essential factor

for real-time respiratory monitoring; as shown in Figure 2g, our triboelectric respiratory sensor works stably after more than 10,000 continuous contact–separation cycles. All these excellent characteristics make this triboelectric respiratory sensor ideal for continuously monitoring daily respiratory status.

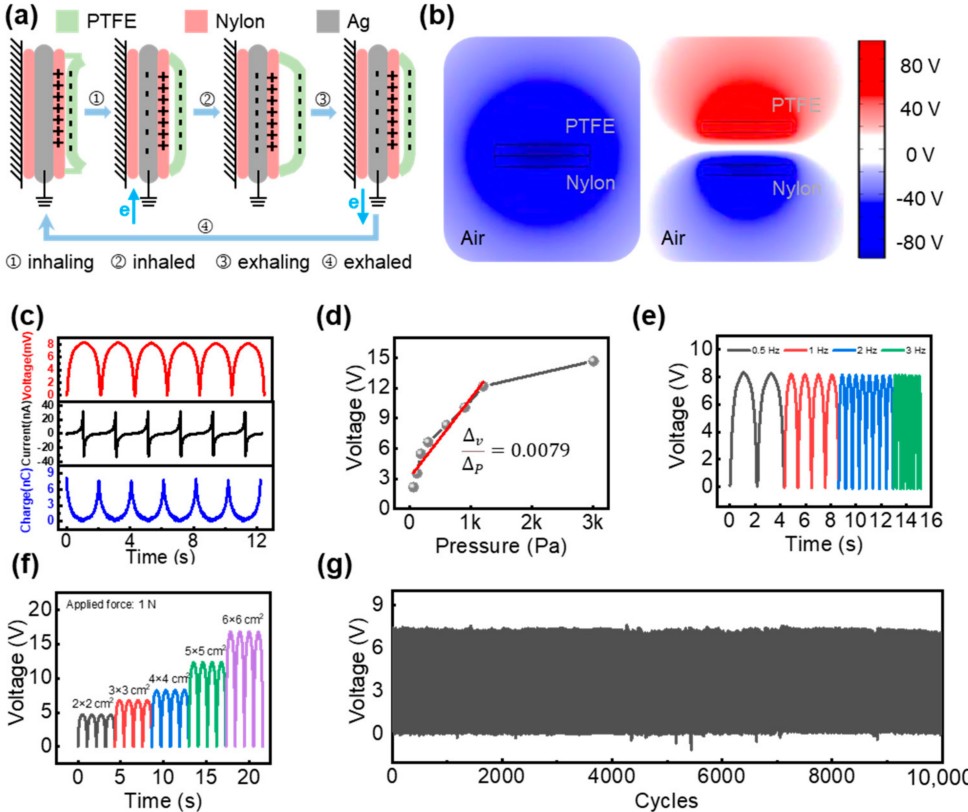

**Figure 2.** The working mechanism and typical output performances of the triboelectric respiratory sensor. (**a,b**) Schematic illustration of the working mechanism and COMSOL finite-element simulation results for the triboelectric respiratory sensor. (**c**) Typical output voltage, current, and charge curves for the triboelectric respiratory sensor. (**d**) Output voltage results for different pressures, from 60 Pa to 3000 Pa. (**e**) Output voltage curves for different operating frequencies, from 0.5 to 3 Hz. (**f**) Output voltage curves for different sensing areas of 2 × 2 to 6 × 6 cm². (**g**). Cycling stability performance of the triboelectric respiratory sensor for 10,000 cycles.

### 3.3. Data Analysis and Feature-Extraction Approaches Respecting Respiratory Signals

Figure 3a–e show the output signal curves of five typical respiratory behaviors recorded by the sensor, including normal breathing, deep breathing, coughing, sneezing, and laughing. These respiratory statuses represent the most common natural physiological behaviors. It can be seen that the electrical signal curves exhibit significantly different waveforms and intensities and provide characteristic information, including but not limited to frequency, intensity, duration, and peak value.

Wavelet transform (Figure 3f–j) and Fourier transform (Figure 3k–o) are used to analyze the electrical signals [26], showing the corresponding time–frequency spectra and frequency–energy distributions, respectively. The wavelet transform and Fourier transform are both powerful tools for analyzing signals. The wavelet transform is useful for analyzing transient non-stationary signals, such as respiratory signals, and can provide a time–frequency map that directly reflects the joint time–frequency characteristics of a signal. The Fourier transform is useful for converting a signal from the time domain to the frequency domain, allowing for the identification of intense frequency bands in the signal. The Fourier transform can also be used to identify the peak energy concentration of a signal, while the wavelet transform can be used to visualize the frequency change band.

Both transforms have their own advantages and can be used to analyze different types of respiratory signals and extract signal features.

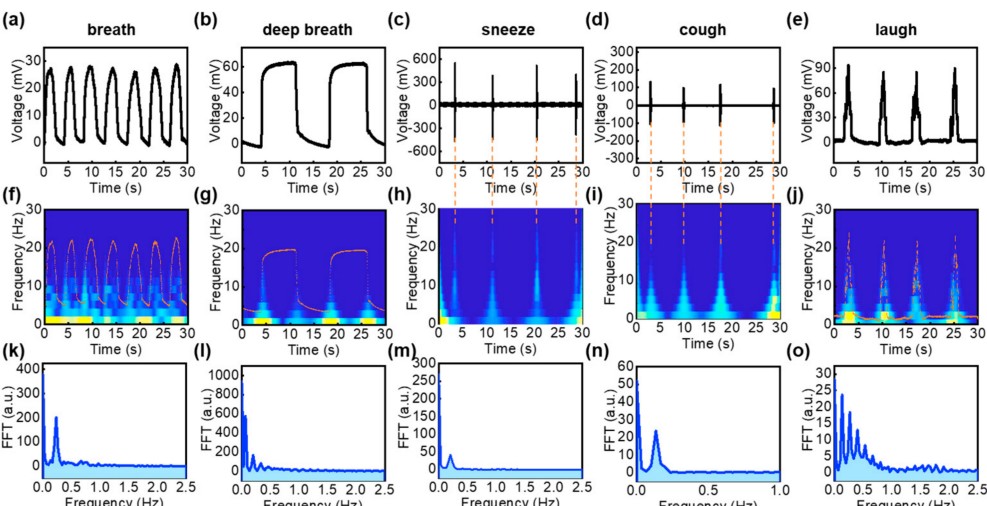

**Figure 3.** Data analysis and feature extraction of five typical respiratory behaviors. (**a–e**) Real-time output voltage curves generated by triboelectric respiratory sensors for normal breathing, deep breathing, sneezing, coughing, and laughing. (**f–j**) The time—frequency spectra corresponding to the five output voltage curves analyzed by wavelet transform. (**k–o**) The corresponding energy distributions of output voltage curves analyzed by Fourier transform.

As shown in Figure 3a,f,k, a peak signal with a voltage of 27.7 mV and a duration of 3.7 s was obtained for the normal breathing process with a 0.24 Hz energy concentration frequency. Deep breathing, which is stronger and longer than normal breathing, meanwhile, presented an output voltage of 63.8 mV, a duration of 14.3 s, and an energy concentration frequency of 0.07 Hz (Figure 3b,g,l). The original curves for coughing and sneezing were similar, but the impact force of the transient airflow generated by sneezing is significantly stronger than that of coughing, which can be verified by the peak distributions of the output voltage and energy concentration curves (Figures 3c,h,m and 3d,i,n). Due to multiple and instantaneous gas-exchange processes, laughing signals present multiple energy concentration peaks mainly concentrated in the range of 0–1 Hz (Figure 3e,j,o).

Finally, a total of 12 typical feature values were extracted from the above time-domain and frequency-domain spectra. From the original time-domain spectra, the mean values, variances, standard deviations, and root mean squares and kurtosis can be extracted. From the frequency-domain spectra, based on the Fourier transform, the features of unbiased estimations, coefficients of variation, and edge factors can be extracted. From the time-domain spectra, based on the wavelet transform, the features of mean values, variances, standard deviations, and root mean squares and kurtosis can be extracted. All these features can be used to analyze the characteristics of respiratory signals and can provide valuable insights into the complex respiratory processes. Based on the abovementioned characteristic information, further training and testing using a machine learning algorithm will enhance the intelligence of the respiratory monitoring system.

### 3.4. Algorithm for the Respiratory Monitoring System and Its Application

Based on a comprehensive understanding of the characteristics of respiratory signals, the decision tree model algorithm is a powerful tool for automatically detecting and classifying complex respiratory behaviors. It has the advantage of reducing over-fitting problems and ranking the importance of extracted features in classification. This makes it an ideal choice for wearable healthcare applications, as it can provide high recognition accuracy.

Figure 4a shows the detailed workflow for the decision tree model algorithm. In the training stage, five classification feature sets are first established based on the extraction of

the above 12 features for each type of respiratory behavior. The structure and flow of the training data set are shown in Figure S9. Using these five classification feature sets and the corresponding annotation labels, we achieved the trained classification model based on the decision tree algorithm. In the test stage, the features of the unknown respiratory signals were extracted and classified one by one according to the trained classification model, until all the features were matched.

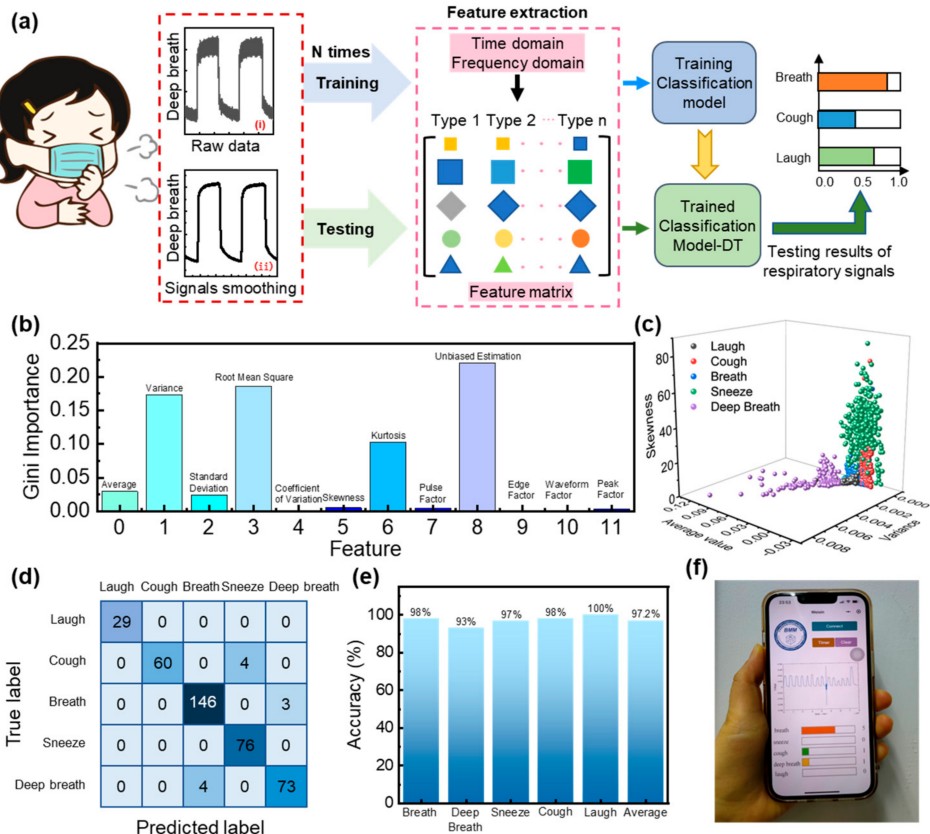

**Figure 4.** Demonstrations of the wireless and real-time respiratory monitoring system. (**a**) The workflow chart of respiratory type recognition based on the decision tree algorithm includes data acquisition, preliminary processing, the algorithm learning process, and the algorithm testing process. (**b**) Gini importance indexes of 12 features in the classification model trained on the 1580 data samples. (**c**) A 3D scatter diagram with average value, variance, and skewness as the features selected in the coordinate axes to divide five respiratory types. (**d**,**e**) Confusion matrix and classification accuracy of the recognition of five respiratory types. (**f**) Optical photograph of the wireless intelligent respiratory monitoring system and a mobile phone app made for real-time respiratory recognition and statistics.

In this work, the number of training data samples for the five types of respiratory signals was 1580. Figure 4b shows the Gini importance indexes of the 12 features in the trained classification model, which are often used to feedback the importance of each feature and judge the contribution in the classification model. It can be seen that variance, root mean square, kurtosis, and unbiased estimation are the most important indicators of the classification model, while coefficient of variance, edge factor, waveform factor, and peak factor contribute less to respiratory behavior classification. We randomly selected 149 normal breathing samples, 77 deep breathing samples, 64 coughing samples, 76 sneezing samples, and 29 laughing samples as a test data set to test the recognition accuracy after training (Figure S10). Such a test method for data extraction is more random and can highlight the accuracy of decision tree model classification. As shown in Figure 4c, a 3D scatter diagram is provided to intuitively highlight the classification ability of the algorithm with respect to the test data set. The signals of sneezing (green) and coughing (red) are

usually difficult to distinguish in waveforms. Here, they can be clearly distinguished by only three selected features, namely, average value, variance, and skewness. According to the confusion matrix, the average accuracy for recognizing respiratory types reached up to 97.2%, indicating the huge potential in terms of meeting daily respiratory monitoring requirements (Figure 4d,e).

Moreover, an intelligent and system-level respiratory monitoring system was demonstrated by integrating the triboelectric respiratory sensor, the portable circuit board (Figure 1e), the decision tree model algorithm, and a mobile terminal. As shown in Figure 4f, the real-time respiratory signals and statistical results for various respiratory behaviors are displayed intuitively in the user interface. This respiratory monitoring system is not only easy to use in daily life but can also monitor a variety of respiratory signals, providing new opportunities for future research on life health monitoring and diagnostic systems.

## 4. Discussion

Our respiratory monitoring system has excellent properties, such as wear resistance, low cost, and high precision, making it a promising technology for early-disease-prediction models, cost-effective daily respiratory care, and treatment-effect tracking. The combination of the triboelectric respiratory sensor and a daily mask produced by the screen-printing method makes the respiratory monitoring system more convenient and comfortable to use. However, our respiratory monitoring system still has some limitations and there is still room for improvement, such as integrating the circuit board and the sensor into a small and light chip and addressing the anchoring deviation of the sensor data due to individual breathing habits and environmental complexity. In addition, further research is needed to deeply analyze the signal characteristics of some specific respiratory diseases based on the mechanical learning algorithm. The ultimate goal of our respiratory monitoring system is to enable high-accuracy medically assisted daily respiratory evaluation through the collection, storage, and objective evaluation of respiratory signals over long-period use times.

## 5. Conclusions

In summary, this paper presents an intelligent, portable, and wireless respiratory monitoring system for real-time evaluation of human respiratory behavior. The respiratory monitoring system consists of a triboelectric respiratory sensor; circuit board hardware for real-time data acquisition, preprocessing, and wireless transmission; a machine learning algorithm to enhance the recognition accuracy of respiratory behaviors; and a mobile terminal app to display the dynamic respiratory curves and statistical data. The triboelectric respiratory sensor—fabricated by the screen-printing method—is thin, lightweight, non-invasive, and made of harmless and odorless materials. It is clearly responsive to the frequency and intensity of respiratory airflow and can easily adjust the electrical output. The portable circuit board is reusable and cost-effective. The decision tree model algorithm is used to identify respiratory signals with an average accuracy of 97.2%. The real-time signal and statistical results can be uploaded to a server network and displayed on various mobile terminals for body health warnings and advice. Thus, this work provides an effective and convenient means of monitoring respiratory health.

**Supplementary Materials:** The following supporting information can be downloaded at: https://www. mdpi.com/article/10.3390/app13063885/s1, Figure S1: Flow chart of the self-powered triboelectric respiratory sensor fabricated by screen printing; Figure S2: SEM images of the screen-printed Ag layer and the nylon layer on the textile; Figure S3: Cross-sectional SEM images of the screen-printed triboelectric respiratory sensor; Figure S4: The weight of the textile printed with the triboelectric respiratory sensor and the pure textile of the same size; Figure S5: Circuit diagram of the self-built portable data-acquisition circuit board; Figure S6: The first response and recovery time of the respiratory sensor for exhalation and inhalation airflow under 0.5 Hz; Figure S7: Output current and charge curves at different operating frequencies, from 0.5 to 3 Hz; Figure S8: The effect of 20–50 °C temperature on the electrical output characteristics of the respiratory sensor; Figure S9: The

training process of the decision tree algorithm; Figure S10: Randomly selected 1 min signal periods of five respiratory signal curves used for the recognition accuracy test of the decision tree algorithm. The test data set included 29 laughing samples, 149 normal breathing samples, 77 deep breathing samples, 64 coughing samples, and 76 sneezing samples. Table S1: Cost of each layer of $4 \times 4$ cm$^2$ respiratory sensor.

**Author Contributions:** C.Z.: Conceptualization, Investigation, Data curation, Methodology, Visualization, Writing—original draft. L.Z.: Conceptualization, Methodology, Formal analysis, Writing—review and editing. Y.T.: Software development, Visualization. B.B.: Software development. D.L.: Supervision. All authors have read and agreed to the published version of the manuscript.

**Funding:** This research was funded by the National Natural Science Foundation of China, grant number 62105238.

**Institutional Review Board Statement:** This study did not involve humans or animals.

**Informed Consent Statement:** Informed consent was obtained from all subjects involved in the study. Written informed consent has been obtained from the patient(s) to publish this paper.

**Data Availability Statement:** No new data were created or analyzed in this study. Data sharing is not applicable to this article.

**Conflicts of Interest:** The authors declare no conflict of interest.

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
