# Peer review of "A Machine-Learning-Algorithm-Assisted Intelligent System for Real-Time Wireless Respiratory Monitoring"

_applsci, doi:10.3390/app13063885_

Round 1
Reviewer 1 Report
The paper presents a mobile health data monitoring system containing a triboelectric respiratory sensor, a hardware circuit board for data acquisition, preprocessing and wireless transmission, a machine learning algorithm to improve recognition accuracy, and a mobile terminal application. The triboelectric sensor - manufactured by the screen printing method features as lightweight, non-invasive and biocompatible. The real-time signal and statistical results can be uploaded to a server network and played on various mobile terminals for body health warnings and advice. There are already some wearable sensors on the market with advantages and disadvantages. See paper two: 10.1002/admt.202100545. I did not detect a great originality, triboelectric sensors with screen printing have been made before. However, not many of them were placed on textiles. The construction is interesting, the materials are biocompatible. The article is well written, you understand what the authors want to convey, the drawings are sufficient and with good resolution. The references are enough, maybe the limitations of the whole system should be written in a few words somewhere in the article, maybe towards the end of the section with results and discussions. I recommend for publication in the current form.
Author Response
Response: We sincerely thank you for your time and effort to review the manuscript. Your feedback and suggestions have been incredibly valuable in enhancing the overall quality of our work. We would also like to thank you for recommending the reference (10.1002/admt.202100545). As a result, we have cited it as Reference #7. In response to your feedback, we have incorporated some sentences in the Discussion Section to discuss the limitations of our respiratory monitoring system. While we have made significant progress, there is still room for improvement, such as integrating the circuit board and sensor into a compact and lightweight chip. Furthermore, we are exploring ways to address the anchoring deviation of the sensor data caused by individual breathing habits and environmental complexity.
[7] Mirjalali, S.; Peng, S.; Fang, Z.; Wang, C.; Wu, S. Wearable Sensors for Remote Health Monitoring: Potential Applications for Early Diagnosis of Covid-19. Adv. Mater. Technol. 2023, 2100545, doi:10.1002/admt.202100545.

Reviewer 2 Report
This paper presents an intelligent, portable, and wireless respiratory monitoring system for real-time evaluation of human respiratory behavior.
in paragraph 2.1, for the material used, the authors cite the names of commercial companies. Is it necessary?? I think the authors should describe the manufacturing of their sensor without citing the names of the companies that sell the components.
in title 3.3, it is “data” analysis and not “Date” analysis
Author Response
Point-by-point Response to reviewers (Manuscript: applsci-2245781).
Reviewers’ comments and point-to-point responses:
Reviewer #2 (Remarks to the author)
This paper presents an intelligent, portable, and wireless respiratory monitoring system for real-time evaluation of human respiratory behavior.
Response: We sincerely thank for your time and effort to review our manuscript. Your comments and suggestions are very valuable and constructive to improve the quality of the manuscript. Our point-by-point responses are provided below.
- In paragraph 2.1, for the material used, the authors cite the names of commercial companies. Is it necessary? I think the authors should describe the manufacturing of their sensor without citing the names of the companies that sell the components.
Response: We sincerely thank the reviewer for carefully reviewing our work in detail. The subtitle of 2.1 is changed from “Fabrication of the triboelectric respiratory sensor” to “Materials and fabrication of the triboelectric respiratory sensor”. The purpose of citing the names of commercial companies is to facilitate other researchers’ reproduction experiments and capable of the mass production.
- In title 3.3, it is “data” analysis and not “Date” analysis.
Response: We sincerely thank the reviewer for carefully reviewing our work in detail and it has been revised.

Reviewer 3 Report
The manuscript proposes a real-time respiration-checking system using a triboelectric sensor, a random forest model, and an integrated circuit board. The results look good; however, there are some issues that the author should address:
1. Reference for each applied module: AD623, ADS1256, BL600-SA-32.
2. The dataset number for each type of typical respiratory behavior is different. Why not collect an equal number of data samples? What effect?
3. Figure 1 shows that the airflow goes in from the nose and out from the mouth. However, we do not use the mouth in normal respiration, can the sensor sense the airflow from the nose?
4. One major problem for a flexible sensor is the connection between the sensor and the circuit board to collect signals. Please explain how to address this problem in the mask in detail.
5. What is the sensitivity parameter of the sensor?
6. What is the sensor response and recovery time?
7. Please provide the thickness of each sensor layer, especially the air gap between the nylon layer and the PTFE layer.
8. What is the structure of the training dataset? How many samples for each behavior?
Author Response
Point-by-point Response to reviewers (Manuscript: applsci-2245781).
Reviewers’ comments and point-to-point responses:
Reviewer #3 (Remarks to the author)
The manuscript proposes a real-time respiration-checking system using a triboelectric sensor, a random forest model, and an integrated circuit board. The results look good; however, there are some issues that the author should address.
Response: We sincerely thank you for your time and effort to review the manuscript. Your comments and suggestions are very valuable and constructive to improve the quality of the manuscript. Our point-by-point responses are provided below.
- Reference for each applied module: AD623, ADS1256, BL600-SA-32.
Response: Thank the reviewer for the careful reviewing working in detail. We have cited the reference #20, #21, and #22 for the applied modules of the AD623 amplifier, ADS1256 analog-to-digital converter, and BL600-SA-32 Bluetooth module in the manuscript, respectively.
[20] Zhang, X.; Wang, J.; Xing, Y.; Li, C. Woven Wearable Electronic Textiles as Self-Powered Intelligent Tribo-Sensors for Activity Monitoring. Glob. chall. 2019, 3, 1900070, doi:10.1002/gch2.201900070.
[21] Xiao, Y.; Shen, D.; Zou, G.; Wu, A.; Liu, L.; Duley, W. W.; Zhou, Y. N. Self-powered, flexible and remote-controlled breath monitor based on TiO2 nanowire networks. Nanotechnology 2019, 30, 325503, doi:10.1088/1361-6528/ab1b93.
[22] Zhang, C.; Zhang, L.; Bao, B.; Ouyang, W.; Chen, W.; Li, Q.; Li, D. Customizing Triboelectric Nanogenerator on Everyday Clothes by Screen‐Printing Technology for Biomechanical Energy Harvesting and Human‐Interactive Applications. Adv. Mater. Technol. 2022, 2201138, doi:10.1002/admt.202201138.
- The dataset number for each type of typical respiratory behavior is different. Why not collect an equal number of data samples? What effect?
Response: We would like to express our appreciation for your thorough review of our work. In the training stage, we sample the same 316 instances for each respiratory behavior. In the testing stage, however, the number of instances for each behavior was randomly extracted by our decision tree algorithm from the total datasets. The randomly extracted dataset instances were utilized to highlight the accuracy of the decision tree model's classification. We also have supplied a more detailed description of this testing method in the manuscript's Results 3.4 section.
- Figure 1 shows that the airflow goes in from the nose and out from the mouth. However, we do not use the mouth in normal respiration, can the sensor sense the airflow from the nose?
Response: We are grateful for the reviewer's insightful questions and constructive feedback. Our respiratory sensor detects both inhalation and exhalation airflow, regardless of whether the breathing is done through the mouth or nose. We appreciate the reviewer for bringing this to our attention, and we hope that our revised manuscript will meet your expectations. Moreover, we have made some modifications to Figure 1a in response to the fact that, in this paper, the inhalation and exhalation airflow is primarily performed through the nose.
Figure 1 (a) Schematic diagram of wearable respiratory monitoring system and its potential expandable medical services.
- One major problem for a flexible sensor is the connection between the sensor and the circuit board to collect signals. Please explain how to address this problem in the mask in detail.
Response: We appreciate your diligent review of our work. Currently, we are using a pluggable DuPont cable to connect the sensor and circuit board, which allows for the easy replacement of masks for daily use and repeatableutilization of the data acquisition circuit boards. The interface between the DuPont cable and sensor is fixed using commercial tape, which works effectively. Furthermore, we plan to further optimize the engineering structure of the insert-press quick wiring terminal for better usability in the future.
- What is the sensitivity parameter of the sensor?
Response: Thank you for your valuable comments. In a static state, adults typically inhale around 500 ml of air per breath, and a respiratory cycle usually lasts about 2.4 seconds (equivalent to 25 breaths per minute). Thus, the average flow rate of a single inhalation or exhalation is approximately 416.7 ml/s. When measured with a gas pressure meter, the gas pressure generated by respiration is approximately 0.9 KPa. We have found that the electrical output of the breathing sensor is almost linear within the range of 60-1200 Pa as shown in Figure 2d, and its sensitivity is 0.0079 V/Pa. The corresponding descriptions have also been supplied in the manuscript.
Figure 2 (d) Output voltage results under the different pressure from 60 Pa to 3000 Pa.
- What is the sensor response and recovery time?
Response: We are grateful for your carefully reviewing our work. The respiratory sensor captures airflow is a continuous and dynamic response process that generates continuous signals over time. According to the definition of the response/recovery time of the gas sensor, that is the time to reach/decrease the (1-1/e) % peak value. After calculation, the response and recovery time of the respiratory sensor are 0.25 s and 0.97 s for 0.5 Hz airflow, respectively (Figure S6). We have added this pertinent information to the manuscript's Results 3.2 section and supporting information.
Figure S6 The first response and recovery time of the respiratory sensor to the exhalation and inhalation airflow under 0.5 Hz.
- Please provide the thickness of each sensor layer, especially the air gap between the nylon layer and the PTFE layer.
Response: We are grateful for the reviewer's professional inquiries and helpful recommendations. The upper PTFE layer measures 20 μm in thickness, and the largest distance between PTFE and Nylon is 1mm. As for the printed component, its top nylon, Ag electrode, bottom nylon, and cotton textile thicknesses are 23 μm, 12 μm, 16 μm, and 500 μm, respectively. The bottom nylon ink is thinner than the top nylon ink due to the penetration into the textile substrate. We have supplied these details in the manuscript's Results 3.1 section.
- What is the structure of the training dataset? How many samples for each behavior?
Response: We appreciate your meticulous examination of our work. To fine-tune the structural parameters of the decision tree model algorithm, we divided the original training data into a training set (80%) and a testing set (20%). Next, we utilized the S-folder cross-validation method and randomly divided the training set (80%) into five parts. Then, we randomly selected one part as the validation set and performed five cross-validations to obtain the optimal respiratory behavior classification model. Finally, we used the divided testing set (20%) to assess the model and ensure that its classification accuracy meets the required standards. Figure S9 was supplied in Supporting Information to depict the training process of the aforementioned decision tree algorithm. We have supplied this information in the manuscript's Results 3.4 section. In addition, the number of each respiratory behavior in the training data set is 316 and the total training data sample is 1580.
Figure S9 The training process of the decision tree algorithm.

Reviewer 4 Report
Dear authors. The manuscript is interesting and presents good results. However, I believe that some points should be clarified before publication.
1-You describe this sensor as Low Cost. However, what would this Low Cost be? What would be the impact on the final value?
2 - You reported sensor stability for 20 min. However, this system will be in use for much longer than that. In this way, as one of the points argued is to be durable, it is important to estimate the useful life of the sensor.
3 - The simulation made in comsol should be better described and discussed. As it stands, it does not provide relevant information.
4 - A point that calls my attention is the fact that breathing alone may not be enough to determine some type of disease. For example, if an increase in breathing rate is observed, how to differentiate through breathing, if the individual is having a problem or simply making a greater effort, such as climbing stairs or doing a light jog? Is it possible to do this just by breathing?
5 – What is the effect of temperature, both of the individual and the environment, on the sensor response?
Author Response
Point-by-point Response to reviewers (Manuscript: applsci-2245781).
Reviewers’ comments and point-to-point responses:
Reviewer #4 (Remarks to the author)
Dear authors. The manuscript is interesting and presents good results. However, I believe that some points should be clarified before publication.
Response: We sincerely thank you for your time and effort to review the manuscript. Your comments and suggestions are very valuable and constructive to improve the quality of the manuscript. Our point-by-point responses are provided below.
- You describe this sensor as Low Cost. However, what would this Low Cost be? What would be the impact on the final value?
Response: We are grateful for the reviewer's insightful and constructive recommendations. Table S1 provides a detailed cost list of our respiration sensor with a 4×4 cm2 area. As a result, a mask suitable for daily respiratory monitoring can be procured for just ~0.377 $, ensuring hassle-free daily replacement without incurring financial strain. This economical respiratory sensor is advantageous for disseminating and popularizing respiratory monitoring systems while alleviating the burden on public healthcare. We have supplied this information into the manuscript's Results 3.1 section and supporting information.
Table S1 Cost of each layer of 4×4 cm2 respiratory sensor
|
Material |
Unit price ($) |
Consumption |
Cost ($) |
Source of purchase |
|
Commercial mask |
0.35/ piece |
1 piece |
0.35 |
www.taobao.com |
|
Nylon ink |
0.011/ml |
0.064 ml |
0.00071 |
Hangzhou MIHE Trading Co. Ltd |
|
Ag ink |
1.09/ml |
0.0192 ml |
0.021 |
Qingdao Nano Print Materials Technology |
|
PTFE film |
2.975/m2 |
16 cm2 |
0.0048 |
Dongguan Hongfu Insulating Material Co., Ltd |
|
Ag adhesive tape |
0.0308/m |
2 cm |
0.00062 |
www.taobao.com |
|
Total |
|
|
0.377 |
|
- You reported sensor stability for 20 min. However, this system will be in use for much longer than that. In this way, as one of the points argued is to be durable, it is important to estimate the useful life of the sensor.
Response: We truly appreciate the reviewer's professional questions and constructive suggestions. We retested the stability of the respiratory sensor for more than 10000 cycles, as shown in Figure 2 On average, adults breathe 20 times per minute or 1200 times per hour. When worn for a maximum of 8 hours, a medical mask will endure over 10,000 respiratory cycles. Our triboelectric respiratory sensor remains stable even after more than 10,000 continuous contact-separation cycles. We have included this information in the manuscript's Results 3.2 section
Figure 2 (g) Cycling stability performance of triboelectric respiratory sensor for 10000 cycles.
- The simulation made in comsol should be better described and discussed. As it stands, it does not provide relevant information.
Response: We truly appreciate the reviewer's professional questions and constructive suggestions. We have incorporated the discussion of simulation outcomes of COMSOL into the manuscript's Results 3.2 section. To obtain a more quantitative understanding of the respiratory sensor's potential distribution, we conducted finite element analysis using COMSOL software, as depicted in Figure 2b. The electrical potential balance is achieved when the PTFE film is in complete contact with the printed nylon layer. Conversely, when the PTFE film is distant from the nylon layer, a potential difference arises between the PTFE film with a positive potential and the nylon layer with a negative potential. The results of the finite element analysis validate our mechanism analysis.
Figure 2 (b) COMSOL finite element simulation results of the triboelectric respiratory sensor.
- A point that calls my attention is the fact that breathing alone may not be enough to determine some type of disease. For example, if an increase in breathing rate is observed, how to differentiate through breathing, if the individual is having a problem or simply making a greater effort, such as climbing stairs or doing a light jog? Is it possible to do this just by breathing?
Response: We appreciate your valuable and constructive feedback, which will serve as our research objective in the upcoming phase. Previous studies (ACS Sens. 2022, 7, 3135-3143) have shown that asthma, bronchitis, chronic obstructive pulmonary disease and general healthy breathing can be effectively distinguished by monitoring respiratory behaviors. For this work, TENG has shown a wide range of applications in bio-mechanical sensing, including motion, pulse, respiration, blinking and voice, as shown in Figure R1. As we prospect in the Discussion, the multi-data fusion and algorithm optimization for respiratory diseases recognition will greatly improve the feasibility and accuracy of determining certain types of diseases.
Figure R1 TENG sensors are widely used in bio-mechanical sensing, including respiratory recognition, motion monitoring, voice recognition, blink sensing, heartbeat monitoring, and gesture recognition.
- What is the effect of temperature, both of the individual and the environment, on the sensor response?
Response: We appreciate your meticulous evaluation of our study. Just as the body temperature is usually around 37 ℃, we measure the electrical output of the respiratory sensor under the temperature range of 20-50 ℃ by a heating platform. As illustrated in Figure S8, we observed that the temperature did not impact the respiratory sensor, thereby guaranteeing its capacity to function effectively in a dynamic environment. We have included this information in the manuscript's Results 3.2 section.
Figure S8 The effect of 20-50 ℃ temperature on the electrical output characteristics of respiratory sensor.

Round 2
Reviewer 3 Report
Thanks for the revision. I recommend publishing this paper.